# Memes: A motif analysis environment in R using tools from the MEME Suite

Spencer L. Nystrom[1,2,3,4], Daniel J. McKay[2,3,4]*

1 Curriculum in Genetics and Molecular Biology, The University of North Carolina at Chapel Hill, Chapel Hill, North Carolina, United States of America, 2 Department of Biology, The University of North Carolina at Chapel Hill, Chapel Hill, North Carolina, United States of America, 3 Department of Genetics, The University of North Carolina at Chapel Hill, Chapel Hill, North Carolina, United States of America, 4 Integrative Program for Biological and Genome Sciences, The University of North Carolina at Chapel Hill, Chapel Hill, North Carolina, United States of America

* dmckay1@email.unc.edu

**Data Availability Statement:** The manuscript source code, raw data, and instructions to reproduce all analysis can be found at github.com/snystrom/memes_paper/ Data used in this manuscript can be found on GEO under

## Abstract

Identification of biopolymer motifs represents a key step in the analysis of biological sequences. The MEME Suite is a widely used toolkit for comprehensive analysis of biopolymer motifs; however, these tools are poorly integrated within popular analysis frameworks like the R/Bioconductor project, creating barriers to their use. Here we present memes, an R package that provides a seamless R interface to a selection of popular MEME Suite tools. memes provides a novel "data aware" interface to these tools, enabling rapid and complex discriminative motif analysis workflows. In addition to interfacing with popular MEME Suite tools, memes leverages existing R/Bioconductor data structures to store the multidimensional data returned by MEME Suite tools for rapid data access and manipulation. Finally, memes provides data visualization capabilities to facilitate communication of results. memes is available as a Bioconductor package at https://bioconductor.org/packages/memes, and the source code can be found at github.com/snystrom/memes.

## Author summary

Biologically active molecules such as DNA, RNA, and proteins are polymers composed of repeated subunits. For example, nucleotides are the subunits of DNA and RNA, and amino acids are the subunits of proteins. Functional properties of biopolymers are determined by short, recurring stretches of subunits known as motifs. Motifs can serve as binding sites between molecules, they can influence the structure of molecules, and they can contribute to enzymatic activities. For these reasons, motif analysis has become a key step in determining the function of biopolymers and in elucidating their roles in biological networks. The MEME Suite is a widely used compilation of tools used for identifying and analyzing motifs found in biological sequences. Here, we describe a new piece of software named "memes" that connects MEME Suite tools to R, the statistical analysis environment. By providing an interface between the MEME Suite and R, memes allows for improved motif analysis workflows and easy access to a wide range of existing data visualization tools, further expanding the utility of MEME Suite tools.

GSE141738 at the following address: https://www.ncbi.nlm.nih.gov/geo/query/acc.cgi?acc=GSE141738.

**Funding:** This work was supported in part by Research Scholar Grant RSG-17-164-01-DDC to D.J.M. from the American Cancer Society (https://www.cancer.org/), and in part by Grant R35-GM128851 to D.J.M. from the National Institute of General Medical Sciences of the NIH (https://www.nigms.nih.gov/). The funders had no role in study design, data collection and analysis, decision to publish, or preparation of the manuscript.

**Competing interests:** The authors have declared that no competing interests exist.

## Introduction

Biopolymers, such as DNA and protein, perform varying functions based on their primary sequence. Short, repeated sequences, or "motifs" represent functional units within biopolymers that can act as interaction surfaces, create structure, or contribute to enzymatic activity. Identification of similar motifs across multiple sequences can provide evidence for shared function, such as identification of kinase substrates based on similarities in phosphorylation site sequence, or characterizing DNA elements based on shared transcription factor binding sequences [1]. Thus the ability to identify, classify, and compare motifs represents a key step in the analysis of biological sequences.

The MEME Suite is a widely utilized set of tools to interrogate motif content across a broad range of biological contexts [2]. With over 25,000 citations to date, and greater than 40,000 unique users of the webserver implementation annually, the MEME Suite has emerged as a standard tool in the field [3,4]. However, several factors limit the full potential of these tools for use in data analysis. MEME Suite tools require carefully formatted inputs to achieve full functionality, yet few tools exist to simplify the process of data formatting, requiring instead that users write custom code to prepare their data, or prepare the inputs by hand, both of which have the potential to be error prone without rigorous testing. Furthermore, the output data from each MEME Suite tool often have complex structures that must be parsed to extract the full suite of information, again requiring users to write custom code for this task. Finally, although the data-generation capabilities of the MEME Suite are excellent, the tools lack powerful ways to visualize the results. Collectively, these factors act as barriers to adoption, preclude deeper analysis of the data, and limit communication of results to the scientific community.

Here we present memes, a motif analysis package that provides a seamless interface to a selection of popular MEME Suite tools within R. Six MEME Suite tools are contained within the memes wrapper (**Table 1**). memes uses base R and Bioconductor data types for data input and output, facilitating better integration with common analysis tools like the tidyverse and other Bioconductor packages [5–7]. Unlike the commandline implementation, memes outputs also function as inputs to other MEME Suite tools, allowing simple construction of motif analysis pipelines without additional data processing steps. Additionally, R/Bioconductor data structures provide a full-featured representation of MEME Suite output data, providing users quick access to all relevant data structures with simple syntax.

**Table 1. A list of MEME Suite tools that are currently included in memes.**

| Tool name | Purpose of tool | Description of output data |
|---|---|---|
| MEME | de novo discovery of ungapped motifs from user-provided sequences. | List of discovered motifs with estimated significance. |
| STREME | de novo discovery of ungapped motifs, optimized for large numbers of sequences. | List of discovered motifs with estimated significance. |
| AME | motif enrichment analysis from user-provided sequences. | List of enriched motifs with estimated significance. |
| Tomtom | matches user-provided motifs to databases of known motifs. | List of matching motifs with estimated significance for the quality of match. |
| FIMO | finds individual occurrences of target motifs in user-provided sequences. | Table of all motif occurrences with estimated significance. |
| DREME | de novo discovery of short ungapped motifs. | List of discovered motifs with estimated significance. |

memes is designed for maximum flexibility and ease of use to allow users to iterate rapidly during analysis. Here we present several examples of how memes allows novel analyses of transcription factor binding profile (ChIP-seq) and open chromatin profile (FAIRE-seq) data by facilitating seamless interoperability between MEME Suite tools and the broader R package landscape.

## Design & implementation

**Core utilities.** MEME Suite tools are run on the commandline and use files stored on-disk as input while returning a series of output files containing varying data types. As a wrapper of MEME tools, memes functions similarly by assigning each supported MEME Suite tool to a run function (runDreme(), runMeme(), runAme(), runFimo(), runTomTom()), which internally writes input objects to files on disk, runs the tool, then imports the data as R objects. These functions accept sequence and motif inputs as required by the tool. Sequence inputs are accepted in BioStrings format, an R/Bioconductor package for storing biopolymer sequence data [8]. Motif inputs are passed as universalmotif objects, another R/Bioconductor package for representing motif matrices and their associated metadata [9]. memes run functions will also accept paths to files on disk, such as fasta files for sequence inputs, and meme format files for motif inputs, reducing the need to read large files into memory. Finally, each run function contains optional function parameters mirroring the commandline arguments provided by each MEME Suite tool. In this way, memes provides a feature-complete interface to the supported MEME Suite tools.

**Output data structures.** MEME tools return HTML reports for each tool that display data in a user-friendly way; however, these files are not ideal for downstream processing. Depending on the tool, data are also returned in tab-separated, or XML format, which are more amenable to computational processing. However, in the case that tab-separated data are returned, results are often incomplete. For example, the TomTom tool, which compares query motifs to a database of known motifs, returns tab-separated results that do not contain the frequency matrix representations of the matched motifs. Instead, users must write custom code to parse these matrices back out from the input databases, creating additional barriers to analysis. In the case that data are returned in XML format, these files contain all relevant data; however, XML files are difficult to parse, again requiring users to write custom code to extract the necessary data. Finally, the data types contained in MEME Suite outputs are context-specific and multidimensional, and thus require special data structures to properly organize the data in memory.

memes provides custom-built data import functions for each supported MEME Suite tool, which import these data as modified R data.frames (described in detail below). These functions can be called directly by users to import data previously generated by the commandline or webserver versions of the MEME Suite for use in R. These import functions also underlie the import step internal to each of the run functions, ensuring consistent performance.

**Structured data.frames hold multidimensional output data.** Data returned from MEME Suite tools are often multidimensional, making them difficult to represent in a simple data structure. For example, MEME and DREME return de-novo discovered motifs from query sequences along with statistical information about their enrichment [10,11]. In this instance, a position frequency matrix (PFM) of the discovered motif can be represented as a matrix, while the properties of that matrix (e.g. the name of the motif, the E-value from the enrichment test, background nucleotide frequency, etc.) must be encoded outside of the matrix, while maintaining their relationship to the corresponding PFM. However, other MEME Suite tools, such as TOMTOM, which compares one motif to a series of several motifs

to identify possible matches, produce an additional layer of information such that input matrices (which contain metadata as previously described) can have a one-to-many relationship with other motif matrices (again with their own metadata) [12]. Thus an ideal representation for these data is one that can hold an unlimited number of motif matrices and their metadata, retain heirarchical relationships, and be easily manipulated using standard analysis tools.

The universalmotif_df data structure is a powerful R/Bioconductor representation for motif matrices and their associated metadata [9]. universalmotif_df format is an alternative representation of universalmotif objects where motifs are stored along rows of a data.frame, and columns store metadata associated with each motif. Adding one-to-many relationships within this structure is trivial, as additional universalmotif_dfs can be nested within each other. These universalmotif_dfs form the basis for a majority of memes data outputs. These structures are also valid input types to memes functions, and when used as such, output data are appended as new columns to the input data, ensuring data provenance. Finally, because universalmotif_dfs are extensions of base R data.frames, they can be manipulated using base R and tidyverse workflows. Therefore, memes data integrate seamlessly with common R workflows.

**Support for genomic range-based data.** Motif analysis is often employed in ChIP-seq analysis, in which data are stored as genomic coordinates rather than sequence. However, MEME Suite tools are designed to work with sequences. While existing tools such as bedtools can extract DNA sequence from genomic coordinates, some MEME tools require fasta headers to be specifically formatted. As a result, users must write custom code to extract the DNA sequence for their genomic ranges of interest.

The memes function get_sequence() automates extraction of DNA sequence from genomic coordinates while simultaneously producing MEME Suite formatted fasta headers. get_sequence() accepts genomic-range based inputs in GenomicRanges format, the *de-facto* standard for genomic coordinate representation in R [13]. Other common genomic coordinate representations, such as bed format, are easily imported as GenomicRanges objects into R using preexisting import functions, meaning memes users do not have to write any custom import functions to work with range-based data using memes. Sequences are returned in Biostrings format and can therefore be used as inputs to all memes commands or as inputs to other R/Bioconductor functions for sequence analysis.

**Data-aware motif analysis.** Discriminative (or "differential") motif analysis, in which motifs are discovered in one set of sequences relative to another set, can uncover biologically relevant motifs associated with membership to distinct categories. For example, during analysis of multiomic data, users can identify different functional categories of genomic regions through integration with orthogonal datasets, such as categorizing transcription factor binding sites by the presence or absence of another factor [14]. Although the MEME Suite allows differential enrichment testing, it does not inherently provide a mechanism for analyzing groups of sequences in parallel, or for performing motif analysis with an understanding of data categories. The memes framework enables "data-aware" motif analysis workflows by allowing named lists of Biostrings objects as input to each function. If using GenomicRanges, users can split peak data on a metadata column using the base R split() function, and then use this result as input to get_sequence(), which will produce a list of BioStrings objects where each entry is named after the data categories from the split column. When a list is used as input to a memes function, it runs the corresponding MEME Suite tool for each object in the list. Users can also pass the name(s) of a category to the control argument to enable differential analysis of the remaining list members against the control category sequences. In this manner, memes enables data-aware differential motif analysis workflows using simple syntax to extend the capabilities of the MEME Suite.

## Data visualization

The MEME Suite provides a small set of data visualizations that have limited customizability. memes leverages the advantages of the R graphics environment to provide a wide range of data visualization options that are highly customizable. We describe two scenarios below.

The TomTom tool allows users to compare unknown motifs to a set of known motifs to identify the best match. Visual inspection of motif comparison data is a key step in assessing the quality and accuracy of a match [12]. The view_tomtom_hits() function allows users to compare query motifs with the list of potential matches as assigned by TomTom, similar to the commandline behavior. The force_best_match() function allows users to reassign the TomTom best match to a lower-ranked (but still statistically significant) match in order to highlight motifs with greater biological relevance (e.g. to skip over a transcription factor that is not expressed in the experimental sample, or when two motifs are matched equally well).

The AME tool searches for enrichment of motifs within a set of experimental sequences relative to a control set [15]. The meaningful result from this tool is the statistical parameter (for example, a p-value) associated with the significance of motif enrichment. However, AME does not provide a mechanism for visualizing these results.

The plot_ame_heatmap() function in memes returns a ggplot2 formatted heatmap of statistical significance of motif enrichment. If AME is used to examine motif content of multiple groups of sequences, the plot_ame_heatmap() function can also return a plot comparing motif significance within multiple groups. Several options exist to customize the heatmap values in order to capture different aspects of the output data. The ame_compare_heatmap_methods() function enables users to compare the distribution of values between samples in order to select a threshold that accurately captures the dynamic range of their results.

## Containerized analysis maximizes availability and facilitates reproducibility

memes relies on a locally installed version of the MEME Suite which is accessed by the user's R process. Although R is available on Windows, Mac, and Linux operating systems, the MEME Suite is incompatible with Windows, limiting its adoption by Windows users. Additional barriers also exist to installing a local copy of the MEME Suite on compatible systems, for example, the MEME Suite relies on several system-level dependencies whose installation can be difficult for novice users. Finally, some tools in the MEME Suite use python to generate shuffled control sequences for analysis, which presents a reproducibility issue as the random number generation algorithm changed between python2.7 and python3. The MEME Suite will build and install on both python2.7 and python3 systems, therefore without careful consideration of this behavior, the same code run on two systems may not produce identical results, even if using the same major version of the MEME Suite. In order to increase access to the MEME Suite on unsupported operating systems, and to facilitate reproducible motif analysis using memes, we have also developed a docker container with a preinstalled version of the MEME Suite along with an R/Bioconductor analysis environment including the most recent version of memes and its dependencies. As new container versions are released, they are version-tagged and stored to ensure reproducibility while allowing updates to the container.

## Results

Here we briefly highlight each of memes current features for analyzing ChIP-seq data. Additional detailed walkthroughs for each supported MEME Suite tool, and a worked example using memes to analyze ChIP-seq data can be found in the memes vignettes and the package

website (snystrom.github.io/memes). In the following example, we reanalyze recent work examining the causal relationship between the binding of the transcription factor E93 to changes in chromatin accessibility during *Drosophila* wing development [14]. Here, we utilize ChIP-seq peak calls for E93 that have been annotated according to the change in chromatin accessibility observed before and after E93 binding. These data are an emblematic example of range-based genomic data (E93 ChIP peaks) containing additional groupings (the chromatin accessibility response following DNA binding) whose membership may be influenced by differential motif content. We show how memes syntax allows sophisticated analysis designs, how memes utilities enable deep interrogation of results, and how memes flexible data structures empower users to integrate the memes workflow with tools offered by other R/Bioconductor packages.

The aforementioned ChIP-seq peaks are stored as a GRanges object with a metadata column (e93_chromatin_response) indicating whether chromatin accessibility tends to increase ("Increasing"), decrease ("Decreasing"), or remain unchanged ("Static") following E93 binding.

```
head(chip_results, 3)
## GRanges object with 3 ranges and 2 metadata columns:
##   seqnames ranges strand | id e93_chromatin_response
##     <Rle> <IRanges> <Rle> | <character> <character>
## [1] chr2L 5651-5750   *  |  peak_1      Static
## [2] chr2L 37478-37577 *  |  peak_3      Increasing
## [3] chr2L 55237-55336 *  |  peak_4      Static
## -------
## seqinfo: 6 sequences from an unspecified genome; no seqlengths
```

Using the get_sequence() function, GRanges objects are converted into DNAStringSet outputs.

```
dm.genome <- BSgenome.Dmelanogaster.UCSC.dm3::BSgenome.Dmelanogaster.
UCSC.dm3
all_sequences <- chip_results %>%
  get_sequence(dm.genome)
head(all_sequences)
## DNAStringSet object of length 6:
##   width seq names
## [1] 100 GGAGTGCCAACATATTGTGCTCT...AAATTGCCGCTAATCAGAAGCAA
chr2L:5651-5750
## [2] 100 CGTTAGAATGGTGCTGTTTGCTG...GCTATGGGACGAAAGTCATCCTC
chr2L:37478-37577
## [3] 100 AAGCAGTGGTCTACGAAACAAAC...CATTCAACATCTGCAAAATCCAG
chr2L:55237-55336
## [4] 100 CAGTACTTCACAACACCTTGCAG...CCGTTTTGTCAACGGGAAACTAG
chr2L:62469-62568
## [5] 100 CGCTGAATTGCACCAAAAGAGCG...AACCCCATCTTTGCATAGTCAGT
chr2L:72503-72602
## [6] 100 CAATCTTACACTCGTTAGATTGC...GCTGTTGCGATGCCATACACTAG
chr2L:79784-79883
```

In order to perform analysis within different groups of peaks, the GRanges object can be split() on a metadata column before input to get_sequence().

```
sequences_by_response <- chip_results %>%
  split(mcols(.)$e93_chromatin_response) %>%
  get_sequence(dm.genome)
```

This produces in a BStringSetList where list members contain a DNAStringSet for each group of sequences.

```
head(sequences_by_response)
## BStringSetList of length 3
```

```
## [["Decreasing"]] chr2L:248266-
248365 = TTAACGAGTGGGGGAGGGAAGAATACGACGAGAGGCGAGG...
## [["Increasing"]] chr2L:37478-
37577 = CGTTAGAATGGTGCTGTTTGCTGTTGGGCGAACGAGGACTAG...
## [["Static"]] chr2L:5651-
5750 = GGAGTGCCAACATATTGTGCTCTACGATTTTTTTGCAACCCAAAATGG...
```

## De novo motif analysis

The DREME tool can be used to discover short, novel motifs present in a set of input sequences relative to a control set. The runDreme() command is the memes interface to the DREME tool. runDreme() syntax enables users to produce complex discriminative analysis designs using intuitive syntax. Examples of possible designs and their syntax are compared below.

```
# Use all sequences vs shuffled background
# Produces:
# − All Sequences vs Shuffled All Sequences
runDreme(all_sequences, control = "shuffle")

# For each response category, discover motifs against a shuffled background set
# Produces:
#  − Increasing vs Shuffled Increasing
#  − Decreasing vs Shuffled Decreasing
#  − Static vs Shuffled Static
runDreme(sequences_by_response, control = "shuffle")

# Search for motifs enriched in the "Increasing" peaks relative to "Decreasing" peaks
# Produces:
# − Increasing vs Decreasing
runDreme(sequences_by_response$Increasing, control = sequences_by_re-
sponse$Decreasing)

# Use the "Static" response category as the control set to discover motifs
# enriched in each remaining category relative to static sites
# Produces:
# − Increasing vs Static
# − Decreasing vs Static
runDreme(sequences_by_response, control = "Static")

# Combine the "Static" and "Increasing" sequences and use as a background set to
# discover motifs enriched in the remaining categories
# Produces:
# − Decreasing vs Static + Increasing
runDreme(sequences_by_response, control = c("Static", "Increasing"))
```

In order to discover motifs associated with dynamic changes to chromatin accessibility, we use open chromatin sites that do not change in accessibility (i.e. "static sites") as the control set to discover motifs enriched in either increasing or decreasing sites.

```
dreme_vs_static <- runDreme(sequences_by_response, control =
"Static")
```

These data are readily visualized using the universalmotif::view_motifs() function. Visualization of the results reveals two distinct motifs associated with decreasing and increasing sites (**Fig 1**). In this analysis, the *de-novo* discovered motifs within each category appear visually similar to each other; however, the MEME Suite does not provide a mechanism to compare groups of motifs based on all pairwise similarity metrics. We utilized the universalmotif::compare_motifs() function to compute Pearson correlation coefficients for each set of motifs to quantitatively assess motif similarity (**Fig 1**).

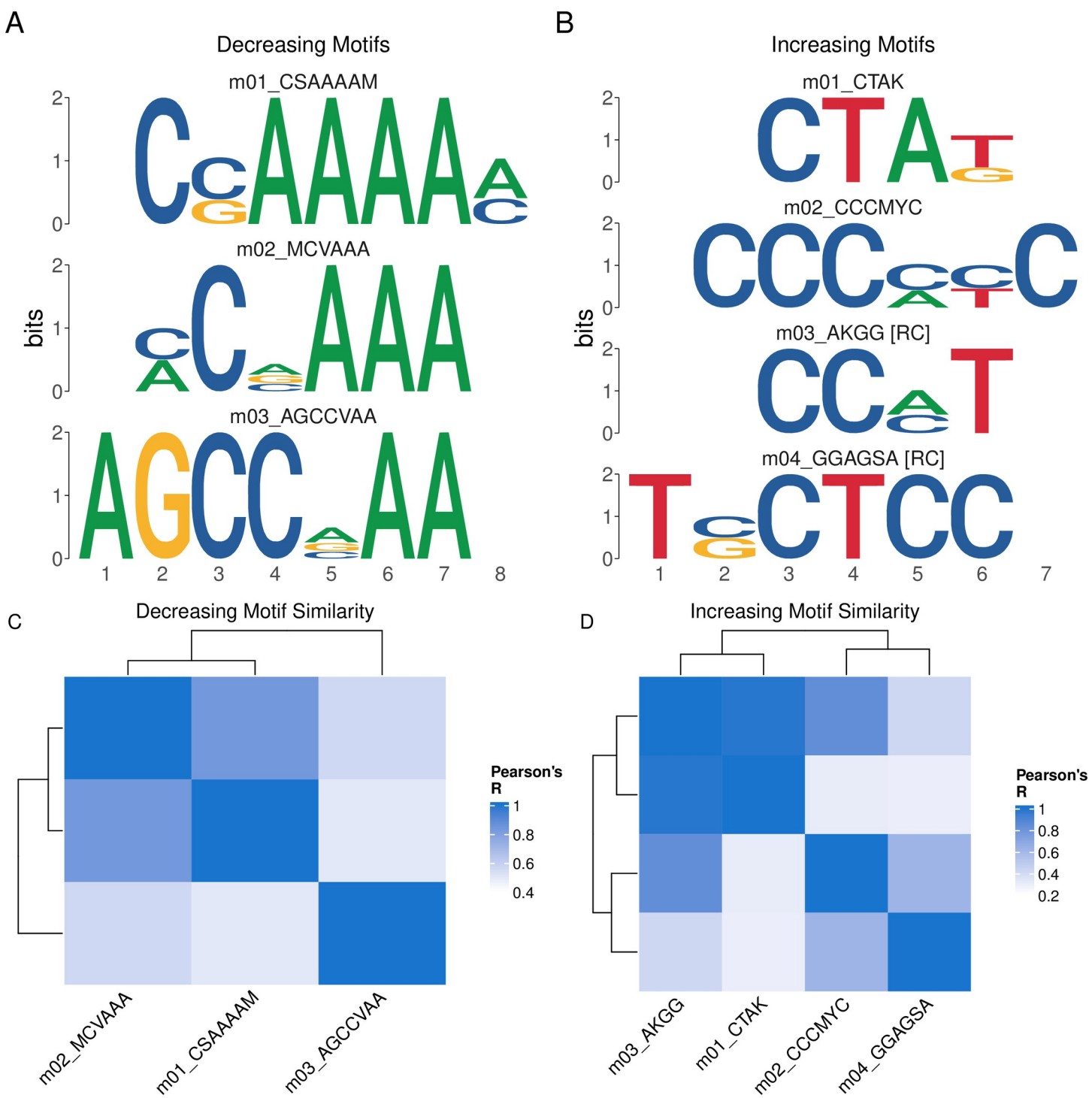

**Fig 1.** PWMs of discovered motifs in Decreasing (A) and Increasing (B) sites. C,D Pearson correlation heatmaps comparing similarity of Decreasing (C) and Increasing (D) motifs.

Transcription factors often bind with sequence specificity to regulate activity of nearby genes. Therefore, comparison of *de-novo* motifs with known DNA binding motifs for transcription factors can be a key step in identifying transcriptional regulators. The TomTom tool

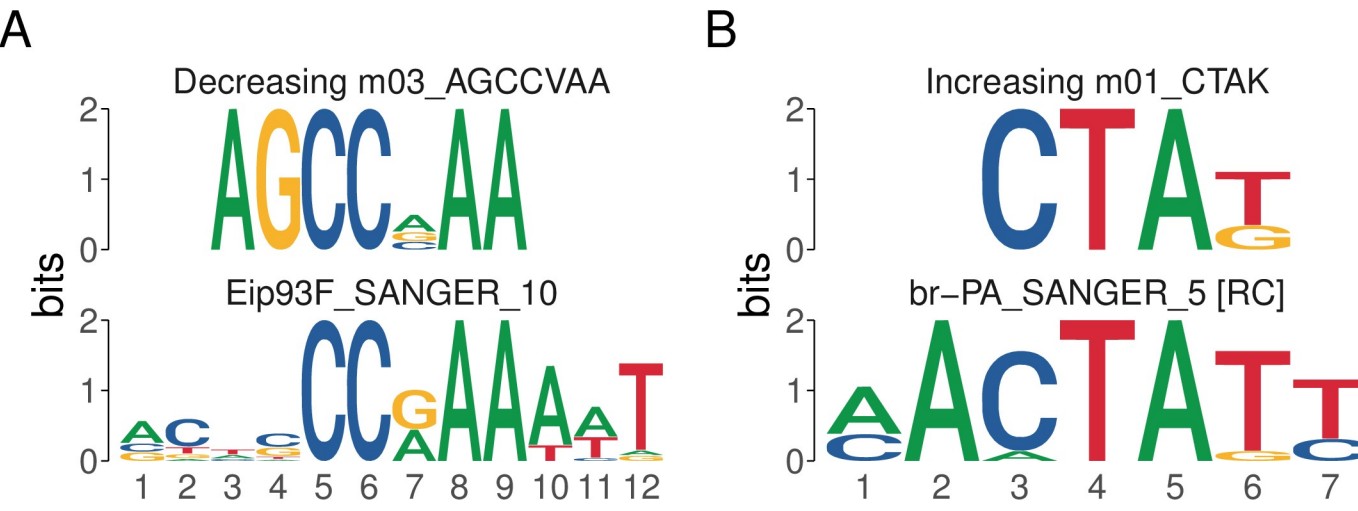

**Fig 2.** Representative plots generated by view_tomtom_hits showing the top hit for motifs discovered in decreasing (A) and increasing (B) peaks.

is used to compare unknown motifs against a list of known motifs to identify matches, memes provides the runTomTom() function as an interface to this tool. By passing the results from runDreme() into runTomTom() and searching within a database of *Drosophila* transcription factor motifs, we can identify candidate transcription factors that may bind the motifs associated with increasing and decreasing chromatin accessibility.

```
dreme_vs_static_tomtom < − runTomTom(dreme_vs_static, database =
"data/flyFactorSurveyCleaned.meme", dist = "ed")
```

Using this approach, the results can be visualized using the view_tomtom_hits() function to visually inspect the matches assigned by TomTom, providing a simple way to assess the quality of the match. A representative plot of these data is shown in **Fig 2**, while the full results can be viewed in **Table 2**. This analysis revealed that the motifs associated with decreasing sites are highly similar to the E93 motif. Conversely, motifs found in increasing sites match the transcription factors Br, L(3)neo38, Gl, and Lola (**Table 2**). These data support the hypothesis that E93 binding to its motif may play a larger role in chromatin closing activity, while binding of the transcription factor br is associated with chromatin opening.

**Table 2. Summary of all TomTom matches for denovo motifs discovered in E93 responsive ChIP peaks.**

| Best match results from TomTom | |
|---|---|
| | Best Match TF |
| Decreasing Motifs | |
| m01_CSAAAAM | CG2052 |
| m02_MCVAAA | Eip93F |
| m03_AGCCVAA | Eip93F |
| Increasing Motifs | |
| m01_CTAK | br |
| m02_CCCMYC | l(3)neo38 |
| m03_AKGG | gl |
| m04_GGAGSA | lola |

## Motif enrichment testing using runAme

Discovery and matching of *de-novo* motifs is only one way to find candidate transcription factors within target sites. Indeed, in many instances, requiring that a motif is recovered *de-novo* is not ideal, as these approaches are less sensitive than targeted searches. Another approach, implemented by the AME tool, is to search for known motif instances in target sequences and test for their over-representation relative to a background set of sequences [15]. The runAme() function is the memes interface to the AME tool. It accepts a set of sequences as input and control sets, and it will perform enrichment testing using a provided motif database for occurrences of each provided motif.

A major limitation of this approach is that transcription factors containing similar families of DNA binding domain often possess highly similar motifs, making it difficult to identify the "true" binding factor associated with an overrepresented motif. Additionally, when searching for matches against a motif database, AME must account for multiple testing, therefore using a larger than necessary motif database can produce a large multiple testing penalty, limiting sensitivity of detection. One way to overcome these limitations is to limit the transcription factor motif database to include only motifs for transcription factors expressed in the sample of interest. Accounting for transcription factor expression during motif analysis has been demonstrated to increase the probability of identifying biologically relevant transcription factor candidates [14,16].

The universalmotif_df structure can be used to integrate expression data with a motif database to remove entries for transcription factors that are not expressed. To do so, we import a *Drosophila* transcription factor motif database generated by the Fly Factor Survey and convert to universalmotif_df format [17]. In this database, the altname column stores the gene symbol.

```
fly_factor <- universalmotif::read_meme("data/flyFactorSurveyCleaned.meme") %
>%
  # Add motif names to the list entry
  setNames(., purrr::map_chr(., ~{.x['name']})) %>%
  to_df()
## # A tibble: 3 x 16
##   name altname family organism consensus alphabet strand icscore
nsites
##   <chr>    <chr> <chr> <chr> <chr>          <chr> <chr> <dbl> <int>
## 1 ab_SANG... ab    <NA>  <NA>  BWNRCCAGGWMCN... DNA    +-    15.0    20
## 2 ab_SOLE... ab    <NA>  <NA>  NNNNHNRCCAGGW... DNA    +-    14.6   446
## 3 Abd-A_F... abd-A <NA>  <NA>  KNMATWAW        DNA    +-     7.37    37
## # ... with 7 more variables: bkgsites <int>, pval <dbl>, qval <dbl>,
eval <dbl>,
## # type <chr>, bkg <named list>, motif <I<named list>>
```

Next, we import a pre-filtered list of genes expressed in a timecourse of *Drosophila* wing development.

```
wing_expressed_genes <- read.csv("data/wing_expressed_genes.csv")
```

Finally, we subset the motif database to only expressed genes using dplyr data.frame subsetting syntax (note that base R subsetting functions operate equally well on these data structures), then convert the data.frame back into universalmotif format using to_list(). This filtering step removes 43% of entries from the original database, greatly reducing the multiple-testing correction.

```
fly_factor_expressed <- fly_factor %>%
  dplyr::filter(altname %in% wing_expressed_genes$symbol) %>%
  to_list()
```

runAme() syntax is identical to runDreme() in that discriminative designs can be constructed by calling list entries by name. Data can be visualized using plot_ame_heatmap(), revealing that the E93 motif is strongly enriched at Decreasing sites, while l(3)neo38, lola, and br motifs are enriched in Increasing sites, supporting the *de-novo* discovery results (**Fig 3**).

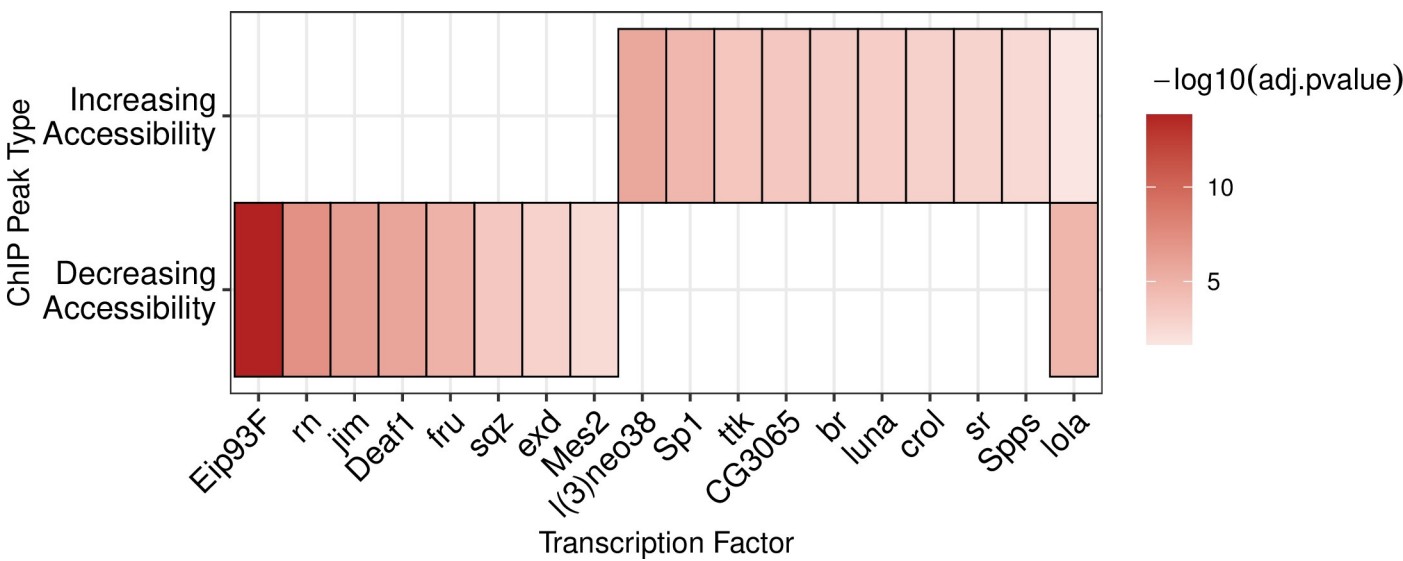

**Fig 3. A heatmap of the -log10(adj.pvalues) from AME enrichment testing within increasing and decreasing sites.**

Additionally, AME detects several other transcription factor motifs that distinguish decreasing and increasing sites, providing additional clues to potential factors that also bind with E93 to affect chromatin accessibility.

```
ame_vs_static <- runAme(sequences_by_response, control = "Static",
database = fly_factor_expressed)
```

**Motif matching using runFimo.** A striking result from these analyses is that the E93 motif is so strongly enriched within E93 ChIP peaks that decrease in accessibility. Significance of motif enrichment can be driven by several factors, such as quality of the query motif relative to the canonical motif or differences in motif number in one group relative to other groups. These questions can be explored directly by identifying motif occurrences in target regions and examining their properties. FIMO allows users to match motifs in input sequences while returning information about the quality of each match in the form of a quantitative score [18].

In order to examine the properties of the E93 motif between different ChIP peaks, we scan all E93 ChIP peaks for matches to the Fly Factor Survey E93 motif using runFimo(). Results are returned as a GRanges object containing the positions of each motif.

```
e93_fimo <- runFimo(all_sequences, fly_factor_expressed$Eip93F_SAN-
GER_10, thresh = 1e-3)
```

Using plyranges, matched motifs can be joined with the metadata of the ChIP peaks with which they intersect [19].

```
e93_fimo %<>%
plyranges::join_overlap_intersect(chip_results)
```

Using this approach we can deeply examine the properties of the E93 motif within each chromatin response category. First, by counting the number of E93 motifs within each category, we demonstrate that Decreasing sites are more likely than increasing or static sites to contain an E93 motif (**Fig 4A**). We extend these observations by rederiving position-weight matrices from sequences matching the E93 motif within each category, allowing visual inspection of motif quality across groups (**Fig 4B**). Notably, differences in quality at base positions 8–12 appear to distinguish increasing from decreasing motifs, where decreasing motifs are more likely to have strong A bases at positions 8 and 9, while E93 motifs from increasing sites are more likely to have a T base pair at position 12 (**Fig 4B**). Examination of bulk FIMO scores (where higher scores represent motifs more similar to the reference) also reveals differences in

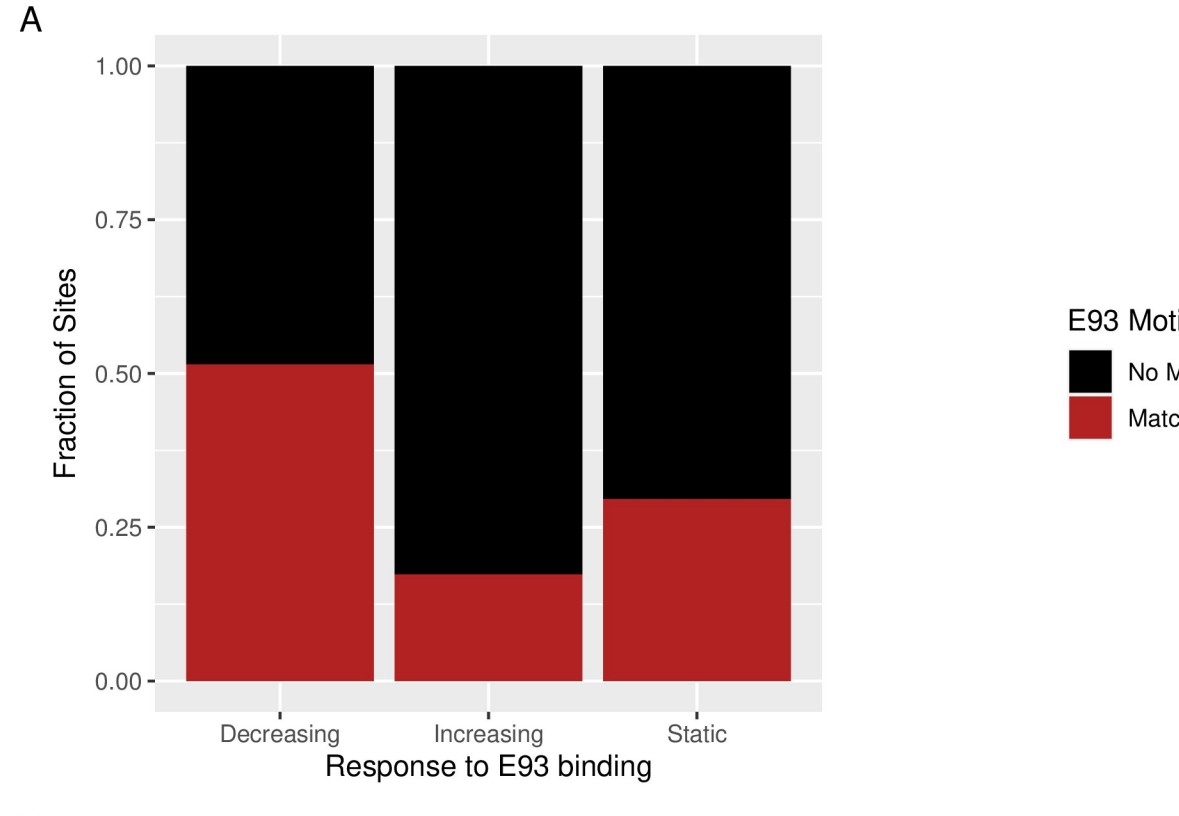

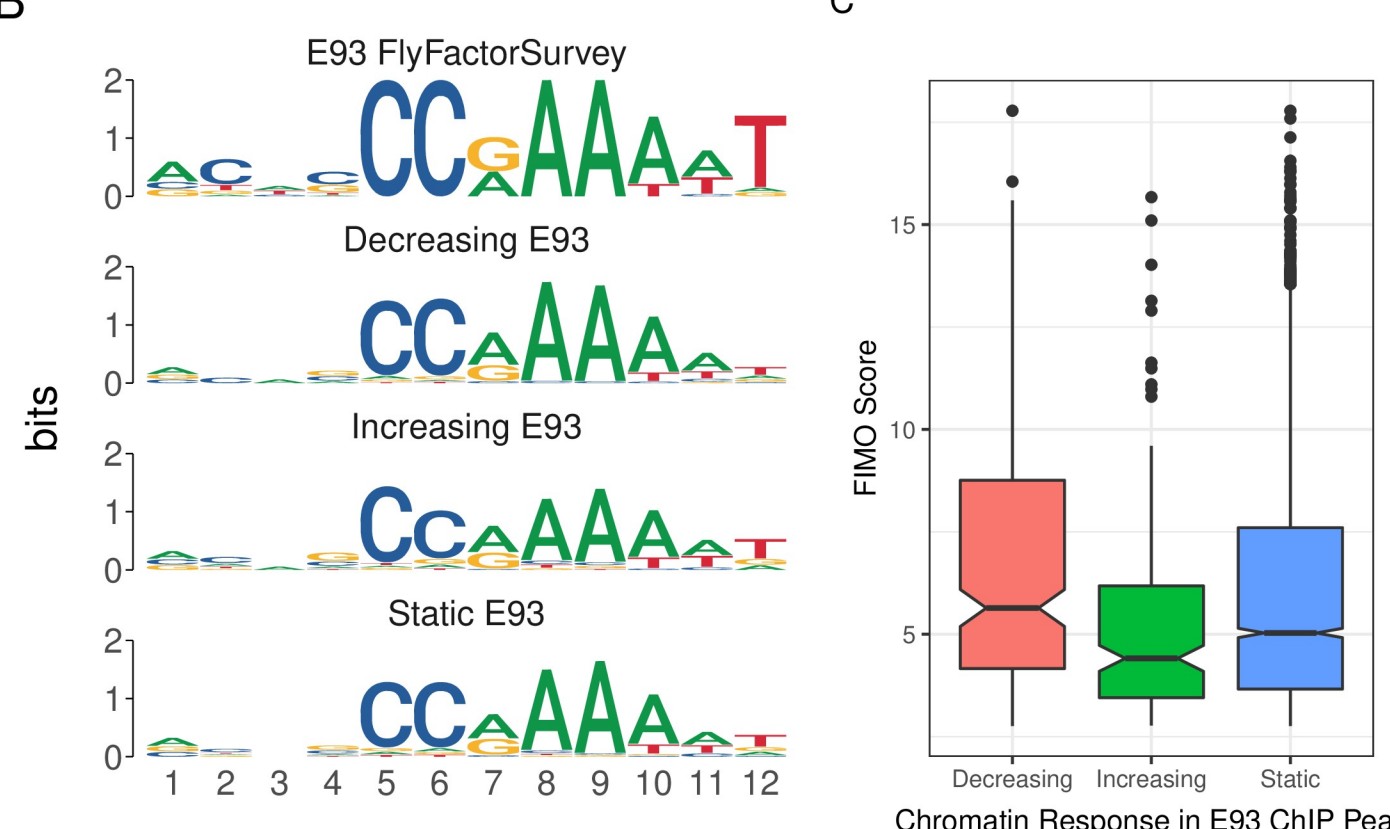

**Fig 4.** A. Stacked barplot showing fraction of each chromatin response category containing E93 motif matches. B. PWMs generated from E93 motif sequences detected in each chromatin response category. C. Boxplot of FIMO Score for each E93 motif within each chromatin response category. Outliers are plotted as distinct points.

E93 motif quality between groups, in particular, E93 motifs from Decreasing sites have higher scores (Fig 4C). Together, these data demonstrate that a key distinguishing factor between whether a site will decrease or increase in chromatin accessibility following E93 binding is the number and quality of E93 motifs at that site.

In summary, memes establishes a powerful motif analysis environment by leveraging the speed and utility of the MEME Suite set of tools in conjunction with the flexible and extensive R/Bioconductor package landscape.

## Availability and future developments

memes is part of Bioconductor. Installation instructions can be found at https://bioconductor.org/packages/memes. The memes package source code is available on github: github.com/snystrom/memes. Documentation is stored in the package vignettes, and also available at the package website: snystrom.github.io/memes. The memes_docker container is available on dockerhub: https://hub.docker.com/r/snystrom/memes_docker, and the container source code is hosted at github: https://github.com/snystrom/memes_docker.

This manuscript was automatically generated using Rmarkdown within the memes_docker container. Its source code, raw data, and instructions to reproduce all analysis can be found at github.com/snystrom/memes_paper/. Data used in this manuscript can be found on GEO at the following accession number: GSE141738.

In the future we hope to add additional data visualizations for examining motif positioning within features. We will continue to add support for additional MEME Suite tools in future versions of the package. Finally, we hope to improve the memes tooling for analyzing amino-acid motifs, which although fully supported by our current framework, may require extra tools that we have not considered.

## Acknowledgments

We would like to acknowledge Megan Justice, Garrett Graham, and members of the McKay lab for helpful comments and feedback. Benjamin Jean-Marie Tremblay added additional features to universalmotif which greatly benefitted the development of memes.

## Author Contributions

**Conceptualization:** Spencer L. Nystrom, Daniel J. McKay.

**Data curation:** Spencer L. Nystrom.

**Formal analysis:** Spencer L. Nystrom.

**Funding acquisition:** Daniel J. McKay.

**Investigation:** Spencer L. Nystrom, Daniel J. McKay.

**Methodology:** Spencer L. Nystrom.

**Project administration:** Daniel J. McKay.

**Resources:** Daniel J. McKay.

**Software:** Spencer L. Nystrom.

**Supervision:** Daniel J. McKay.

**Validation:** Spencer L. Nystrom, Daniel J. McKay.

**Visualization:** Spencer L. Nystrom, Daniel J. McKay.

**Writing – original draft:** Spencer L. Nystrom, Daniel J. McKay.

**Writing – review & editing:** Spencer L. Nystrom, Daniel J. McKay.

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
