## [Decision Letter · Decision Letter 0]

14 Jun 2021

Dear Dr. McKay,

Thank you very much for submitting your manuscript "Memes: an R interface to the MEME Suite" for consideration at PLOS Computational Biology.

As with all papers reviewed by the journal, your manuscript was reviewed by members of the editorial board and by several independent reviewers. In light of the reviews (below this email), we would like to invite the resubmission of a significantly-revised version that takes into account the reviewers' comments.

I would like the authors to especially address the following issue raised by one of the reviewers: "memes only provide an R interface to 35% of the MEME toolset, which does not make memes a R/Bioconductor wrapper for the whole MEME Suite". 

We cannot make any decision about publication until we have seen the revised manuscript and your response to the reviewers' comments. Your revised manuscript is also likely to be sent to reviewers for further evaluation.

Sincerely,

Mihaela Pertea

Software Editor

PLOS Computational Biology

Mihaela Pertea

Software Editor

PLOS Computational Biology

Reviewer's Responses to Questions

**Comments to the Authors:**

Reviewer #1: This manuscript by Nystrom and MacKay describes “memes”, an R/Bioconductor package that provides an interface to the MEME suite of tools for analysis of sequence motifs. While the package largely acts as a wrapper around existing MEME suite tools, the functions it provides will make it easy to incorporate these tools into an R-based reproducible analysis pipeline, including publication-quality visualisations thanks to the new visualisation functions included in the package. In addition, the fact that memes uses Bioconductor data structures to provide a familiar interface will make these tools accessible to a wide range of users by reducing the need for users to write custom code to format their data. Overall, the package should be a very useful tool to the many users of the MEME suite and Bioconductor.

I have only minor comments:

1. I would find it helpful to have a figure or table early in the paper with an overview of the tools available in the MEME suite, which of these are already implemented in memes, their purposes (i.e., de novo motif identification, motif enrichment, motif matching, etc), and perhaps a visual example of the output. There are many options available in the MEME suite/memes, so providing an overview early on would help give context and guide the reader through the rest of the manuscript.

2. Although most of the software and R/Bioconductor packages that are mentioned have references, there are no references included for R itself, the Bioconductor project, or the tidyverse packages. Given the tight integration with Bioconductor, the authors should consider adding a citation for at least the Bioconductor project.

Reviewer #2: In this paper, Nystrom et al. presented a R/Bioconductor wrapper (memes) for MEME Suite. MEME Suite is a well-known and well-maintained collection of tools (command-line and web-server) to discover and analyze motifs using DNA, RNA, and protein sequences. There are 17 different tools available in MEME Suite, from motif discovery and motif enrichment to motif comparison.

Although the authors did a great job by providing memes for the MEME toolset; however, it provides the interfacing to only 35% (6 out of 17) of the tools available in the MEME suite. Hence the title and abstract are misleading. Additionally, the authors provide some useful visualization functions, for example, to compare motifs analysis across conditions.

While some may argue that we do not need a wrapper in the first place for a tool that is available as a command-line and as web-interface for users with less computational skills. Nevertheless, it is good to provide efficient wrappers in programming languages with a large user base (such as Python and R) for well-used and maintained tools, given the reason is well justified. However, on the flip side often the citation credit goes to the wrapper tool, not the actual tool that runs in the background and does the heavy lifting. But the authors in their code documentation are listing and asking memes users to cite relevant MEME tools, which is highly appreciated.

The manuscript is adequately written with use cases, but the wording in the title and main text needs attention and re-phrasing. Throughout the manuscript, the authors used the word “complex” to describe MEME output, structure, and analysis. I have been a MEME user for several years and did not find it complex - but user friendly and easy to run and interpret the results. The six wrappers are nicely documented and easy to use in R. The source code is available on GitHub, Bioconductor, and also through a docker container.

Given running the MEME tools in the background and importing the output to R is trivial, which authors claim “complex” throughout the manuscript (as many of these tools provide text/xml outputs). Further, memes only provide an R interface to 35% of the MEME toolset, which does not make memes a R/Bioconductor wrapper for the whole MEME Suite. I think the current version of the manuscript and the software are not in shape for a standalone publication.

Reviewer #3: The authors propose an R library for accessing MEME suite functionality. They provide a user with the instruments to convert their data in the required format and to work with MEME suite output. The package seems to be well-designed and is adapted to work with other common Bioconductor packages. It passes all CRAN and Bioconductor strict package requirements. In general, I see no reason why this package should not be published

**Have the authors made all data and (if applicable) computational code underlying the findings in their manuscript fully available?**

Reviewer #1: Yes

Reviewer #2: Yes

Reviewer #3: Yes

PLOS authors have the option to publish the peer review history of their article (what does this mean?). If published, this will include your full peer review and any attached files.

Reviewer #1: No

Reviewer #2: No

Reviewer #3: **Yes: **Dmitry Penzar
---

## [Decision Letter · Decision Letter 1]

10 Sep 2021

Dear Dr. McKay,

We are pleased to inform you that your manuscript 'Memes: a motif analysis environment in R using tools from the MEME Suite' has been provisionally accepted for publication in PLOS Computational Biology.

Best regards,

Mihaela Pertea

Software Editor

PLOS Computational Biology

Feilim Mac Gabhann

Editor-in-Chief

PLOS Computational Biology

Reviewer's Responses to Questions

**Comments to the Authors:**

Reviewer #1: Thank you to the authors for adding the overview table and additional citations. I am satisfied that the revised manuscript fully addresses my previous comments and those of the other reviewers.

Reviewer #2: In this revised version the authors addressed most of my concerns. I do not have further comments.

**Have the authors made all data and (if applicable) computational code underlying the findings in their manuscript fully available?**

Reviewer #1: Yes

Reviewer #2: Yes

PLOS authors have the option to publish the peer review history of their article (what does this mean?). If published, this will include your full peer review and any attached files.

Reviewer #1: **Yes: **Elizabeth Ing-Simmons

Reviewer #2: No

---

## [Editor Report · Acceptance letter]

20 Sep 2021

PCOMPBIOL-D-21-00641R1 

Memes: a motif analysis environment in R using tools from the MEME Suite

Dear Dr McKay,

I am pleased to inform you that your manuscript has been formally accepted for publication in PLOS Computational Biology. Your manuscript is now with our production department and you will be notified of the publication date in due course.

With kind regards,

Andrea Szabo
